SOFTWARE

# scRepertoire 2: Enhanced and efficient toolkit for single-cell immune profiling

Qile Yang[1], Ksenia R. Safina[2,3], Kieu Diem Quynh Nguyen[4], Zewen Kelvin Tuong[4], Nicholas Borcherding[5]*

1 University of California Berkeley, Berkeley, California, United States of America, 2 Division of Hematology, Brigham and Women's Hospital, Boston, Massachusetts, United States of America, 3 Broad Institute of MIT and Harvard, Cambridge, Massachusetts, United States of America, 4 Ian Frazer Centre for Children's Immunotherapy Research, Child Health Research Centre, Faculty of Health, Medicine and Behavioural Sciences, The University of Queensland, Brisbane, Australia, 5 Department of Pathology and Immunology, Washington University School of Medicine, Saint Louis, Missouri, United States of America

* borcherding.n@wustl.edu

## Abstract

Single-cell adaptive immune receptor repertoire sequencing (scAIRR-seq) and single-cell RNA sequencing (scRNA-seq) provide a transformative approach to profiling immune responses at unprecedented resolution across diverse pathophysiologic contexts. This work presents scRepertoire 2, a substantial update to our R package for analyzing and visualizing single-cell immune receptor data. This new version introduces an array of features designed to enhance both the depth and breadth of immune receptor analysis, including improved workflows for clonotype tracking, repertoire diversity metrics, and novel visualization modules that facilitate longitudinal and comparative studies. Additionally, scRepertoire 2 offers seamless integration with contemporary single-cell analysis frameworks like Seurat and SingleCellExperiment, allowing users to conduct end-to-end single-cell immune profiling with transcriptomic data. Performance optimizations in scRepertoire 2 resulted in a 85.1% increase in speed and a 91.9% reduction in memory usage from the first version over the range repertoire size tested in benchmarking, addressing the demands of the ever-increasing size and scale of single-cell studies. This release marks an advancement in single cell immunogenomics, equipping researchers with a robust toolset to uncover immune dynamics in health and disease.

## Introduction

High-throughput sequencing technologies are foundational to modern biomedical innovation, driving advancements across various fields. Single-cell technologies stand out for their unparalleled ability to dissect cellular heterogeneity and dynamics across tissues and conditions, making them indispensable in immunology and oncology research. By combining scRNA-seq with scAIRR-seq, derived either through

**Data availability statement:** scRepertoire is built within the R framework and is freely available under the MIT license. It is hosted on Bioconductor (https://bioconductor.org/packages/scRepertoire). Comprehensive documentation, tutorials, and source code are provided to support users at https://github.com/ncborcherding/scRepertoire. The code and detailed methods for reproducing analysis are available at https://github.com/BorchLab/scRepertoire.v2_manuscript.

**Funding:** This work was supported by internal departmental funding from the Washington University Department of Pathology and Immunology (https://pathology.wustl.edu/) to NB. The funders had no role in study design, data collection and analysis, decision to publish, or preparation of the manuscript' statement.

**Competing interests:** We have read the journal's policy and the authors of this manuscript have the following competing interests: Q.Y. was previously employed by Generation Lab, Inc. N.B. was previously employed by Santa Ana Bio and Omniscope and is currently an advisor to Epana Bio and consultant for Columbus Instruments. The work presented does not pertain to any commercial endeavors in the companies listed above.

direct sequencing or inference from transcriptomic data, researchers can concurrently analyze gene expression and immune receptor diversity at the single-cell level [1]. The capability to profile both the transcriptional states and clonotype structures of T and B cells through scRNA and scAIRR data provides a robust framework for tracking immune cell activation, clonal expansion, and persistence. These are critical parameters for assessing vaccine efficacy, evaluating immune responses in cancer, responses in cellular therapies, like chimeric antigen receptor T-cell therapies, and elucidating mechanisms underlying autoimmune diseases [2].

Despite the rapid growth of scAIRR analysis, limitations remain with existing tools (reviewed in S1 Table), primarily available in Python and R [3]. Many tools lack robust integration for immune receptor profiling with transcriptomic data or flexibility in data export formats, which hinders reproducibility and cross-platform compatibility in expanding datasets [3]. The increasing demand for standardized, flexible, and scalable analysis frameworks underscores the need for improved software solutions in this field. The growing size and scope of single-cell data creation also underscores this need for improved standardization and efficiency.

The initial release of scRepertoire addressed some of these needs, establishing itself as an integral tool within single-cell workflows due to its compatibility with widely used R-based single-cell analysis platforms like Seurat and SingleCellExperiment [4]. scRepertoire enabled researchers to merge, filter, and visualize clonotype data, offering essential resources for the in-depth characterization of immune repertoires at the single-cell level. As one of the first open-source packages for scAIRR analysis, scRepertoire has over 32,000 downloads from Bioconductor and is a broad utility for the field. Nevertheless, much like the fast-paced changes in scRNA analysis, scRepertoire required a significant rebuild to support new and emerging analyses and increasing sizes of experimental data sets.

Here, we present the latest release of scRepertoire, which introduces a suite of powerful new features and enhancements aimed at refining immune receptor analysis and visualization and are the focus of the Design and Implementation section. This update includes advanced tools for summarizing amino acid sequences, such as positional entropy and residue-specific composition profiles, alongside visualizations of Kmer distributions and gene pairings. User utilities have been optimized for speed, with faster generation of clonal pairs (S1 Fig), improved clustering, and expanded support for importing and exporting clone data across data types. Additional features include identifying T and B cell receptor (TCR/BCR) doublets and improved repertoire comparisons with customizable normalization. scRepertoire seamlessly interacts with deep learning modules such as Trex [5], Ibex [6], and ImmApex [7] and other packages, such as APackOfTheClones [8], scPlotter [9], and DandelionR [10], facilitating a more comprehensive and scalable approach to single-cell immune profiling.

## Design and implementation

### Workflow

The scRepertoire package provides a comprehensive, R-based framework for immune repertoire analysis, seamlessly integrating clonotype data with transcriptomic profiles to enable sophisticated insights into immune cell populations (Fig 1). The recent redesign of scRepertoire emphasizes accessibility and usability, with universalized function names and language to make the package approachable for novice and experienced users. Documentation has been significantly expanded, and a dedicated pkgdown website now hosts extensive examples tutorials, and documentation, creating a readily accessible knowledge base for users to maximize the package's capabilities. A full summary of the changes is available in S2 Table.

This workflow processes data from multiple sequencing and alignment formats (e.g., TSV, JSON, CSV) and is compatible with outputs from widely used alignment pipelines, supporting flexibility in data input. Following data import,

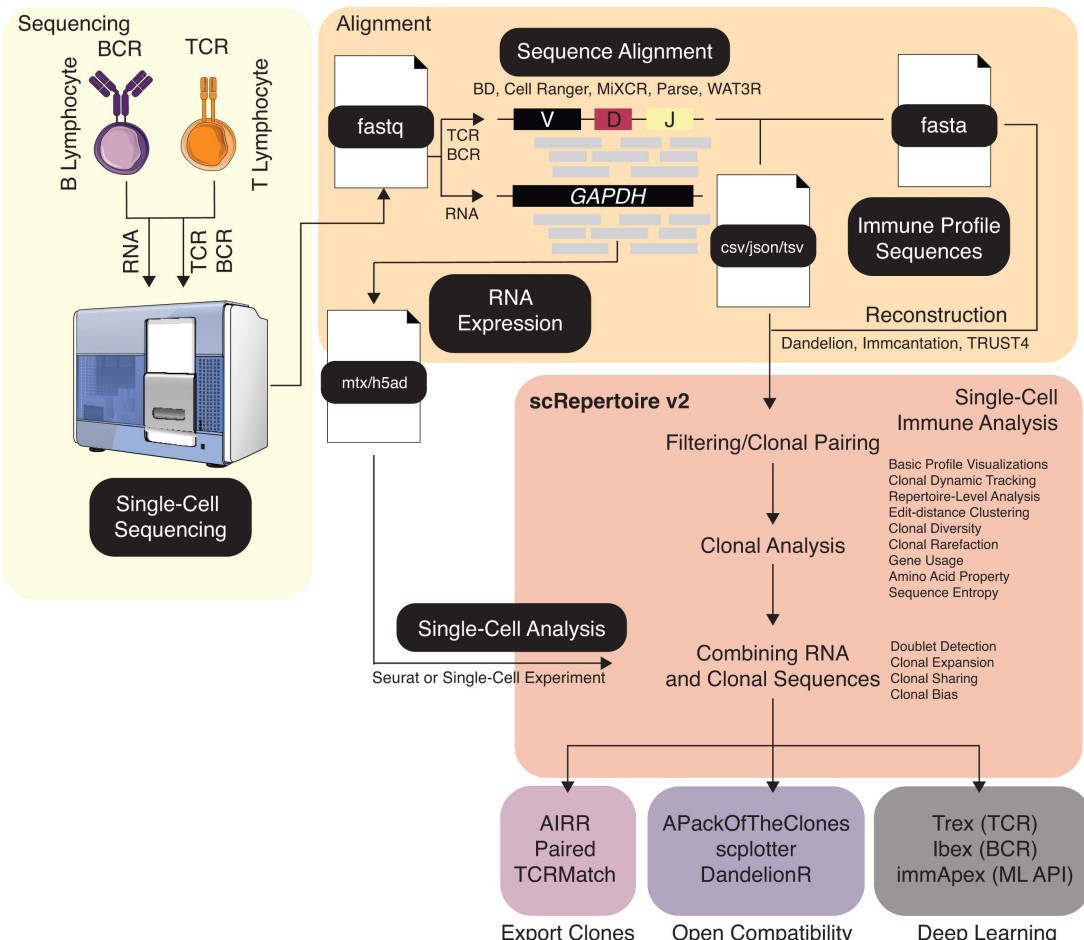

**Fig 1. Schematic overview of the scRepertoire workflow and analyses.** Sequence alignments from standard pipelines or custom reconstruction methods are directly imported into scRepertoire, which performs filtering and pairing of B- or T-lymphocyte receptors. A variety of clonal analyses and visualizations are then available, with the resulting immune receptor data seamlessly attachable to single-cell RNA, protein, or chromatin assays processed via Seurat or SingleCellExperiment. scRepertoire also offers native compatibility with immApex, Trex, and Ibex, enabling deep-learning–based embeddings of scAIRR sequences for classifier development or multimodal data integration. The figure utilizes images from the NIAID BioArt (https://bioart.niaid.nih.gov/) repository under the CC-BY license.

scRepertoire performs stringent clonal pairing and quality control to assign clonotypes at the single-cell level. The assignment of paired clones allows for robust clonal analysis that quantifies and visualizes clonal expansion and lineage relationships within B-cell and T-cell populations. By linking RNA expression data with clonal sequences, scRepertoire provides an integrative view of immune clonotypes within their transcriptional context, which is essential for uncovering immune cells' functional states and diversity, such as analyzing pathogenic sequences in autoimmunity [11] (S2 Fig) or vaccination responses [5] (S3 Fig). In addition to core analysis capabilities, scRepertoire supports downstream applications through compatible tools: Trex, Ibex, and immApex, which aid in the development of models and classifiers for the TCR, BCR, or adaptive immune receptors, respectively. These extensions allow users to tailor their analyses toward specific receptor types and leverage advanced computational techniques, including machine learning, for predictive modeling.

## Expanded data compatibility

scRepertoire 2 supports a range of scAIRR-seq formats, including 10x Genomics, AIRR, BD Rhapsody, MiXCR [12], Parse Bio Evercode, TRUST4 [13], and WAT3R [14]. With the additional format support, a new data importer *loadContigs()* has been introduced and automatically detects input formats, loading scAIRR-seq data from either a list in the R environment or a directory. In addition, the loadContigs() has robust error handling, warning users of misclassified formats and allowing users to assert the format directly. It should be noted that scRepertoire functions on aligned data and is dependent on the proper preprocessing and alignment of the supported formats. Users need to understand the data being imported into scRepertoire. For exporting data, we added *exportClones()*, a flexible solution for exporting scRepertoire clonal data, making it compatible with multiple formats, such as AIRR or TCRmatch [15], for easy integration into downstream analysis tools. This function allows users to save tabular clone information organized by barcodes and sequence details such as those compatible with AIRR, supporting seamless data sharing and interoperability across pipelines. In addition, scRepertoire now allows custom clonal definitions in a cell-wise metadata column so that clones identified with alternative pipelines can be visualized/analyzed within scRepertoire.

## Performance optimizations

With the integration of C++ source code via Rcpp, essential methods have been re-engineered internally to significantly reduce both practical runtime overhead and theoretical time complexity. The most important enhanced methods include those for generating clonal pairs for both with *combineTCR()* and *combineBCR()*, where the runtime of the receptor sequence - cell pairing step now approximates 85.1% faster with a 91.9% reduction in memory utilization from the previous version over a range of 1000 to $1 \times 10^6$ cells in the benchmarking analysis (S1 Fig). In the current version, scRepertoire v2 can process $1 \times 10^6$ cells in a median time of 32.9 seconds, faster than other pipelines tested (S1 Fig). For BCR clonotype inference and general clonal clustering based on Levenshtein distance, the runtime has also been reduced to scale linearly. In addition, threshold length filtering has been implemented when clustering clones by edit distance. A similar optimization was performed to calculate amino acid and nucleotide-based Kmer counts. For all visualization functions, we implemented a conditional evaluation to prioritize the order of operations based on the user's specified output. Specifically, if the user selects data export and plotting functionalities, the package now evaluates and executes data export tasks before initiating plotting.

## Enhanced repertoire summarization

The latest release of scRepertoire introduces advanced features for comprehensive immune repertoire summarization, focusing on amino acid composition and variable, diversity, and joining (VDJ) gene usage. The *positionalProperty()* function facilitates the examination of physical properties along the complementarity-determining region 3 (CDR3) sequence, enabling detailed analysis of specific sequence regions potentially involved in antigen specificity or structural stability.

Additionally, the *positionalEntropy()* function quantifies variability along CDR3 sequences by measuring entropy at each amino acid residue, allowing for identifying conserved or highly variable motifs that may have implications for epitope recognition. Further augmenting these capabilities, the suite of amino acid compositional tools enhances the analysis of amino acid and sequence motifs. The *percentAA()* function calculates amino acid composition frequencies across receptor sequences, supporting the detection of conserved amino acid signatures across clonotypes. Similarly, *percentKmer()* evaluates Kmer distribution, providing a granular assessment of recurring sequence motifs across diverse clonotypes or experimental conditions. Finally, *percentVJ()* visualizes the frequency and distribution of V-J gene pairings in an intuitive heatmap format, facilitating the identification of overrepresented gene combinations within immune repertoires.

## Clonal diversity analysis

The *clonalRarefaction()* function in scRepertoire offers a versatile framework for rarefaction analysis, allowing users to estimate clonal richness while accounting for potential sampling biases. Importantly, the function also computes statistical uncertainties by deriving confidence intervals via bootstrap resampling. This feature quantifies the variability inherent in diversity estimates due to sampling differences, thereby strengthening the reliability of cross-sample comparisons. This capability is especially valuable in comparative studies of immune responses across diverse experimental conditions, where controlled adjustments for sampling depth and uncertainty are essential. Moreover, the function's grouping feature further enhances its utility by enabling structured comparisons across experimental or biological conditions, supporting a robust and nuanced assessment of clonal diversity under varied settings. Additionally, scRepertoire has streamlined the *StartracDiversity()* function, removing custom data dependencies to enhance computational efficiency and accessibility. This re-implementation facilitates estimating differential clonal expansion, cross-tissue migration, and state transitions using differential diversity quantifications, offering critical insights into clonal dynamics within and between tissue environments.

## Empowering ML applications

Building on the established Seurat and Bioconductor SingleCellExperiment infrastructure, our VDJ integration builds a direct bridge to structured data for machine/deep learning. An application example built upon scRepertoire 2 is the amino acid autoencoders Trex [5] and Ibex, which can leverage scAIRR sequences to create meaningful latent dimensions. These latent dimensions can be used independently or incorporated into multimodal single-cell analysis workflows and have been used to perform antigen prediction with in vitro validation [5]. Both models use ImmApex [5], a scRepertoire-compatible companion toolkit that facilitates deep and machine learning for immune receptor sequences for classification or predictive modeling. This allows researchers to perform supervised or unsupervised analysis of their own or public data.

## Results

An example data set for exploring scRepertoire capabilities is provided by Jiang et al. [13]. This data set includes single-cell RNA and immune profiling (both TCR and BCR) of erythema migrans lesions (SKL) and adjacent normal skin (SKN), offering a unique view into immune activity in distinct tissue microenvironments. The cohort consists of six distinct patients. However, two samples were removed for the downstream analyses due to the lack of scAIRR sequences or lack of sequencing of a secondary tissue. After quality control filtering, individual lymphocytes with productive scAIRR sequences were retained, identifying 13 distinct cell types (Fig 2A). These cell types demonstrated varied clonal expansion profiles, providing insights into immune cell dynamics within the tissue (Fig 2B).

Plasma cells exhibited the most prominent clonal expansion among the identified cell types. In the 192567SKL sample, a single clone accounted for over 50% of the cluster, highlighting its dominant presence (Fig 2B). Other cell types with

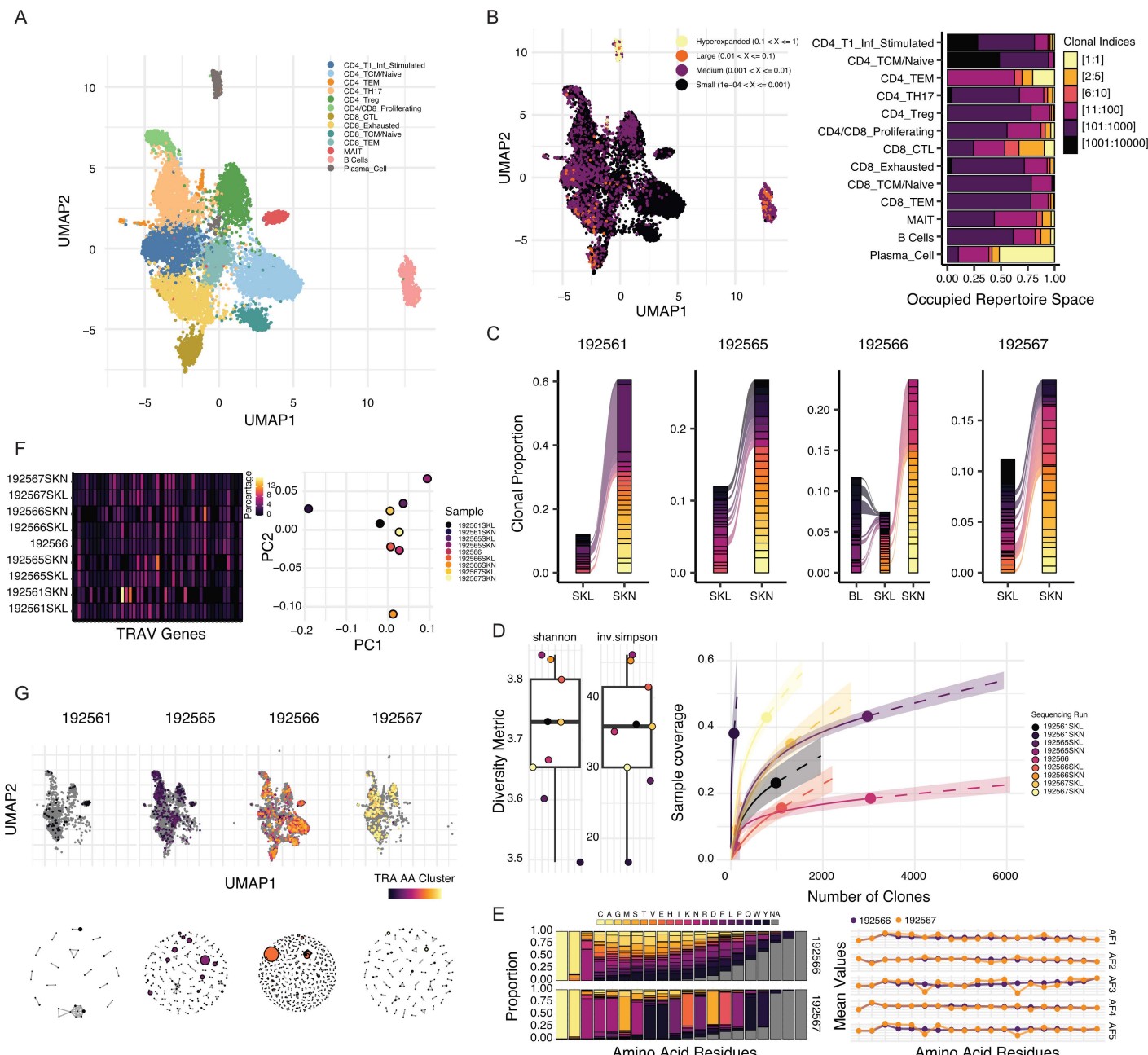

**Fig 2. Single-cell analysis of lymphocytes in erythema migrans lesions and adjacent normal skin. A.** UMAP projection derived from single-cell RNA and TCR/BCR sequencing, revealing distinct lymphocyte subpopulations. **B.** UMAP overlay highlighting clonal expansion by relative frequency. Bar plots illustrate occupied clonal space, ranked by clone size and stratified by lymphocyte cell type. **C.** Alluvial plots showing TCR clones shared among erythema migrans lesions (SKL), adjacent normal skin (SKN), and blood (BL), featuring the top 20 clones per patient by repertoire proportion. **D.** Boot-strapped Shannon and inverse Simpson diversity values for each sequencing run with clonal rarefaction and extrapolation curves based on Shannon diversity indices. **E.** Proportion of amino acids and mean Atchley factor values across the heavy-chain CDR3 region for samples 192566 and 192567. **F.** Heatmap of TRAV gene usage alongside principal component analysis of sequencing runs. **G.** TRA chain clustering based on amino acid sequences using a normalized Levenshtein edit distance threshold (0.85), visualized on a UMAP and via a Fruchterman–Reingold layout in igraph. The dot size indicates the number of sequences or extent of clonal expansion.

significant expansion included CD4 + tissue effector memory (CD4_TEM) cells, cytotoxic CD8 + T (CD8_CTL) cells, and mucosal-associated invariant T (MAIT) cells. These findings underscore the diversity of clonal activity across immune cell populations.

To further explore immune dynamics, the top 20 TCR clones were examined for clonal sharing across tissues within the same patient (Fig 2C). Notably, overlap was observed between SKN and SKL samples, suggesting potential roles for tissue-resident T cells in both environments. In the 192566 sample, which included sequencing from peripheral blood (BL), top TCR clones shared between BL and SKL samples were mutually exclusive of SKN samples. This finding suggests specific trafficking or retention mechanisms. Interestingly, no BCR clonal sharing was observed across tissues, potentially reflecting different modes of immune compartmentalization or low overall sampling.

To quantify repertoire diversity, scRepertoire applied Shannon diversity and inverse Simpson diversity estimates (Fig 2D). By bootstrapping across the smallest repertoire in the comparison group and averaging results from 100 runs, scRepertoire provided robust diversity estimates. Additionally, diversity metrics enabled interpolation and extrapolation using the iNEXT package [14], facilitating rarefaction and sample completeness assessments. These analyses revealed differences in diversity that may correspond to distinct immune pressures or clonal selection processes.

Repertoire summarization within scRepertoire provided additional insights into amino acid use and sequence properties. For the two samples with BCR sequences within multiple tissues, CDR3 sequence analysis revealed a predominant usage of serine (S) at positions 6 and 11 in sample 192567 (Fig 2E). Analysis of amino acid properties, such as Atchley factors (AF) [16], indicated positional decreases in molecular size (AF3) and electrostatic charge (AF5) associated with these serine-enriched positions, potentially influencing antigen-binding characteristics.

Additional summarization approaches included calculating the percentage of amino acid Kmers, VJ gene pairing, and single gene usage within scAIRR chains (Fig 2F). These metrics provided a comprehensive overview of repertoire composition. Dimensional reduction techniques, such as principal component analysis, were applied to exported data (Fig 2F). Clustering of clones by nucleotide or amino acid sequence, based on normalized Levenshtein distance (using 0.85 as a threshold), can enable the identification of potential antigen-driven selection pressures (Fig 2G). Clustering could be performed across all sequences or grouped by categorical variables, such as patient samples. Moreover, scRepertoire supports exporting clustering networks as igraph objects [17], facilitating integration with network-science-based analytical tools.

These analyses illustrate the breadth of scRepertoire's capabilities for immune repertoire analysis. Additional analyses across several pathological states are available in S2 and S3 Figs. By integrating sequence-level insights with functional and diversity metrics, scRepertoire provides a robust framework for studying immune responses in complex tissue environments.

## Availability and future directions

scRepertoire is built within the R framework and is freely available under the MIT license. It is hosted on Bioconductor (https://bioconductor.org/packages/scRepertoire). Comprehensive documentation, tutorials, and source code are provided to support users at https://github.com/BorchLab/scRepertoire. The code and detailed methods for reproducing the erythema migrans and supplemental analyses are available at https://github.com/BorchLab/scRepertoire.v2_manuscript.

Future development plans for scRepertoire focus on enhancing its interoperability with other immune receptor analysis tools. Improved compatibility will encourage custom extensions and collaborative tool development, fostering the growth of related packages such as ApackOfTheClones [8] and DandelionR [10]. These integrations aim to streamline workflows for users analyzing complex single-cell immune repertoires.

Another key development area is incorporating advanced statistical methods and machine learning applications within scRepertoire and its compatible extensions. These enhancements will allow the uncovering of deeper insights into

immune repertoire dynamics, predict immune responses, and identify biomarkers for clinical and research purposes. Many of these areas are active in terms of research and development (reviewed in [18]). In the long term, scRepertoire's framework should be extended to accommodate multi-omics integration. For example, combining immune repertoire data with spatial transcriptomics or proteomics could provide a more comprehensive picture of immune responses, such as the interaction of adaptive immune cells with the microenvironment (spatial) or high throughput antigen discovery (proteomics).

## Supporting information

**S1 Table. Summary of capabilities of scAIRR-seq analysis software.** Check marks equate to supported functionality for single-cell sequences. PHYLO/SHM, Phylogenetic and somatic hypermutation. *Immunarch supports PHYLO/SHM for non-single-cell formats. **VDJVIEW has feature differentiation for BCRs versus TCRs, however no apparent method for calling BCR clonotypes for BCRs beyond strict sequence-based methods.
(DOCX)

**S2 Table. Comprehensive summary of addition and changes to exported functions in scRepertoire v2.**
(DOCX)

**S1 Fig. Benchmarking execution time and memory usage across single-cell immune profiling pipelines. A.** Median execution time (in seconds) across 10 independent runs for each pipeline, evaluated on datasets ranging in size from 2000 to 1,024,000 cells. **B.** Median memory usage (in megabytes) across 10 independent runs for each pipeline, evaluated on the same dataset sizes. Pipelines include *immunarch*, *vdjdj*, *scRepertoire v1 [1]*, and *scRepertoire v2*.
(EPS)

**S2 Fig. Analysis of paired PBMC and aqueous humor sampling in uveitis patient cohort. A.** Schematic overview showing the number of enrolled patients (N = 18) and the total T-cell count (112,255) after filtering for both gene expression and TCR sequences. **B.** UMAP projection of all T cells, colored by broad T-cell subset annotations. **C.** Normalized mRNA expression of canonical T-cell markers overlaid on the UMAP. **D.** UMAP highlighting clonal expansion, with colors reflecting each clone's relative abundance. **E.** Identification and distribution of YEIH$^{232-240}$ T cells visualized on the UMAP (top) and quantified by cluster (bottom), illustrating enrichment of these putative pathogenic T cells in clusters 3 and 13. **F.** STARTRAC metrics for cross-tissue migration (migr), state transitions (trans), and clonal expansion (expa), shown by cluster.
(EPS)

**S3 Fig. Temporal profiling of T-cell responses in the draining lymph nodes following COVID19 mRNA vaccination. A.** UMAP projection of 34,846 T cells from a single HLA-DPB1*04-restricted individual at four time points - days 28 (second vaccine dose), 60, 110, and 201 - post-vaccination. **B.** Visualization (top) and quantification by cluster (bottom) of Spike$^{167-180}$ -specific T cells (n = 168). **C.** Minimal clonal sharing of Spike$^{167-180}$ -specific T cells across the four time points, shown by proportional distribution of each clone. **D.** Comparison of amino acid usage in the TCRA CDR3 region among Spike$^{167-180}$ -specific clones versus other T cells in the draining lymph node. **F.** Principal component analysis of TCRA sequences embedded using Trex for all 34,846 T cells, separated by time point, with Spike$^{167-180}$ -specific clones highlighted in color. The figure utilizes images from the NIAID BioArt (https://bioart.niaid.nih.gov/) repository under the CC-BY license.
(EPS)

**S1 Text. Supplemental methods for benchmarking analysis.**
(DOCX)

## Acknowledgments

From the time of original publishing to scRepertoire 2, the community that uses scRepertoire has contributed consistently to improving the package. We would like to specifically acknowledge the users that directly contributed code to the package: Massimo Andreatta, Nicholas Bormann, Scott Nicholas Furlan, Gloria Kraus, I-Hsuan Lin, Jacqueline HY Siu, Kelvin Tuong, and Panwen Wang. Also, we thank GitHub users Simon-Leonard and Liripo. In addition, we would like to thank Jonathan Noonan and Kyle Romine their constructive suggestions for scRepertoire improvements. We also thank Philip Mudd and Michael Paley for the valuable insights into their data sets that appear in the supplemental figures.

## Author contributions

**Conceptualization:** Qile Yang, Nicholas Borcherding.

**Formal analysis:** Ksenia R. Safina, Kieu Diem Quynh Nguyen, Zewen Kelvin Tuong, Nicholas Borcherding.

**Investigation:** Ksenia R. Safina, Nicholas Borcherding.

**Methodology:** Qile Yang, Nicholas Borcherding.

**Software:** Qile Yang, Ksenia R. Safina, Nicholas Borcherding.

**Supervision:** Zewen Kelvin Tuong, Nicholas Borcherding.

**Validation:** Kieu Diem Quynh Nguyen, Zewen Kelvin Tuong, Nicholas Borcherding.

**Visualization:** Qile Yang, Nicholas Borcherding.

**Writing – original draft:** Qile Yang, Nicholas Borcherding.

**Writing – review & editing:** Zewen Kelvin Tuong, Nicholas Borcherding.

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
