## [Decision Letter · Decision Letter 0]

PCOMPBIOL-D-24-02249

scRepertoire 2: Enhanced and Efficient Toolkit for Single-Cell Immune Profiling

PLOS Computational Biology

Dear Dr. Borcherding,

Thank you for submitting your manuscript to PLOS Computational Biology. After careful consideration, we feel that it has merit but does not fully meet PLOS Computational Biology's publication criteria as it currently stands. Therefore, we invite you to submit a revised version of the manuscript that addresses the points raised during the review process.

Please submit your revised manuscript within 30 days May 23 2025 11:59PM. If you will need more time than this to complete your revisions, please reply to this message or contact the journal office at ploscompbiol@plos.org. Please include the following items when submitting your revised manuscript:

We look forward to receiving your revised manuscript.

Kind regards,

Pramod Shinde

Guest Editor

PLOS Computational Biology

Amber Smith

Section Editor

PLOS Computational Biology

**Additional Editor Comments :**

The manuscript on scRepertoire 2 presents a significant update to the existing toolkit, with improved computational efficiency, expanded data format compatibility, and integration with machine learning frameworks. The new features and performance enhancements add value to single-cell immune repertoire analysis, and the erythema migrans case study effectively demonstrates the toolkit’s capabilities. However, the reviewers have raised several critical issues that need to be addressed to strengthen the manuscript and broaden its impact. They emphasized the need for more rigorous benchmarking against existing tools to provide a clearer comparative analysis of performance and functionality. The machine learning integration, while promising, would benefit from a more concrete example to show how it can generate meaningful insights. Reviewers also suggested clarifying statistical uncertainty in diversity analysis and expanding the results section to better illustrate how the toolkit enhances biological interpretation. Improvements in presentation and clarity, such as defining acronyms at first use, providing more detailed figure legends, strengthening references, and ensuring overall consistency, were also recommended. I suggest major revisions to address all the key points raised by the reviewers and resubmit after thoroughly incorporating the feedback.

**Journal Requirements:**

3) We noticed that you used the phrase 'data not shown' in the manuscript. We do not allow these references, as the PLOS data access policy requires that all data be either published with the manuscript or made available in a publicly accessible database. Please amend the supplementary material to include the referenced data or remove the references.

Potential Copyright Issues:

i) Figure 1. Please confirm whether you drew the images / clip-art within the figure panels by hand. If you did not draw the images, please provide (a) a link to the source of the images or icons and their license / terms of use; or (b) written permission from the copyright holder to publish the images or icons under our CC BY 4.0 license. Alternatively, you may replace the images with open source alternatives. See these open source resources you may use to replace images / clip-art:

ii) The following Figure contains a logo or branding: 1. We are not permitted to publish this under our CC-BY 4.0 license, even with permission. We ask that you please remove or replace it.

**Reviewers' comments:**

Reviewer's Responses to Questions

Reviewer #1: Exploring a fresh version of scRepertoire is a significant and valuable contribution. It is necessary to improve and expand the evaluation and presentation of results and performance.

The abstract is quite generic, so the authors could include summary statistics to assess the performance of their method and support their assertions in the abstract.

Please explain the acronyms before starting to use them. For instance, what does 'scAIRR-seq' mean?

You should add multiple references to support your statements. For instance, this sentence requires multiple references: "Many tools lack robust integration for immune receptor profiling with transcriptomic data or flexibility in data export formats, which hinders reproducibility and cross-platform compatibility in expanding datasets."

It is necessary to create a table and compare the functionality of your package with those of the published packages, including scRepertoire v1.

The aim of scAIRR-seq and Single Cell Immune Profiling is to provide comprehensive insights into the immune system at the single-cell level. However, their scope and integration of different data modalities have nuances. You need to discuss this in your article. Can your tool be used for Single Cell Immune Profiling?

Can your tool help improve the antigen specificity information of scAIRR-seq?

The method section focuses mainly on reviewing functions. It would be helpful if you could give more details about the methodology behind each function to help understand their functionality and as a guide for picking the appropriate one for downstream analysis. Additionally, it is important to highlight any new features in your package. Is it simply a collection of previously released tools or do you include additional components to address the shortcomings in the field?

The results section will become more extensive by presenting examples of how this new tool enhances biological interpretation. Can you provide a demonstration of biological insights that can be achieved using ScRepertoire?

I am unable to find a discussion section. Is it not necessary to include a discussion section?

Reviewer #2: The manuscript presents scRepertoire 2, a major update to an R-based toolkit for single-cell adaptive immune receptor repertoire sequencing (scAIRR) analysis. The package optimizes computational efficiency, expands data format compatibility, introduces novel repertoire analysis features, and integrates with deep learning frameworks. A case study on erythema migrans lesion samples demonstrates its practical application, though validation across multiple disease contexts would strengthen the work.

Major Strengths

Substantial performance gains via C++ integration, reducing computational complexity from quadratic to linear scaling

Expanded data format support with automatic format detection, eliminating the need for file conversion

Novel analytical tools (positional entropy, amino acid property analysis, k-mer distributions) enable deeper repertoire characterization

Enhanced visualization modules and improved metrics for repertoire diversity analysis

Seamless integration with machine learning frameworks supports predictive modeling in computational immunology

The erythema migrans case study effectively demonstrates the package's capabilities

Major Weaknesses

No benchmarking against competing tools to quantitatively demonstrate efficiency gains

Limited validation based on a single dataset—additional disease models would improve generalizability

Insufficient technical details on algorithmic implementations, particularly for novel features

Lack of error case analysis—no discussion of potential failure scenarios

Minimal validation of new metrics across diverse datasets

Recommendations

Include benchmarking results comparing performance against tools like Immcantation, scirpy, and TRUST4

Expand dataset validation using scAIRR data from cancer immunotherapy, autoimmune diseases, or vaccine studies

Provide detailed algorithmic descriptions for novel features

Discuss potential failure cases (e.g., low cell counts, dropout effects, misaligned assignments)

Enhance comparison with alternative tools to contextualize unique contributions

Include runtime efficiency comparisons on large-scale datasets

Explore future directions such as spatial transcriptomics integration and multi-omics approaches

The manuscript makes a significant contribution to single-cell immunogenomics. The performance improvements, expanded analytical capabilities, and deep learning integration are valuable for immune profiling research. However, addressing the benchmarking gaps, dataset limitations, and validation concerns would substantially strengthen the work.

Reviewer #3: In the manuscript of Yang et al, the authors present an updated version of the single-cell immune repertoire analysis tool, scRepertoire2, with a strong emphasis on integrating diverse single-cell data formats for upstream processing and incorporating various amino acid autoencoders in the downstream analyses. In the updated package, the enhancements in immune repertoire summarization, with the focus on amino acid composition and gene usage, have the potential to drive further developments in the field. Overall, the manuscript is well-constructed and I recommend acceptance of the manuscript after minor revisions.

1. Authors demonstrate the updated scRepertoire2 by analyzing transcriptomic and immune repertoire profiling data from erythema lesions. Authors did a good job in detailing the new features. More quantitative benchmarking with other packages would help highlight the unique advantages of scRepertoire2.

2. One unique improvement of scRepertoire 2 is this compatibility with ML packages (like Trex, Ibex and ImmApex). Although the authors have included vignettes in the online tutorials, it would be helpful is author could include a more detailed example in the manuscript, demonstrating how scRepertoire 2 can integrate with those packages and what unique biological insights can be acquired.

3. Single-cell dataset volumes are increasing dramatically. For example, ParseBio has published several immune profiling datasets with over 1 million cells. Are there specific optimizations for large-scale dataset(like disk storage or memory management) implemented in this version? How does scRepertoire2 compatilble with large-scale data (>1 miilions cells)? Authors should provide the information on scalability and performance with very large datasets.

4. In the future direction section, authors should discuss alternative approaches currently addressing similar problems and how further development of scRepertoire2 would fit within this landscape.

5. Please ensure all figures have clear legends and all abbreviations should be well-defined.

Reviewer #4: The study by Yang et. al. presents scRepertoire 2, a computational R package designed for immune repertoire analysis. The tool integrates single-cell RNA-seq and TCR/BCR sequencing data, enabling users to characterize immune clonotypes in their transcriptomic context. The key strengths of this work include:

- Enhanced usability & accessibility: The redesign introduces intuitive function names, improved documentation, and a pkgdown website to support users.

- Expanded data compatibility: Supports multiple sequencing pipelines (e.g., 10x Genomics, AIRR, MiXCR, TRUST4), improving adaptability.

- Performance optimizations: Efficient C++ integration reduces computational complexity, allowing linear scalability for key functions.

- Comprehensive repertoire summarization: Advanced features such as entropy analysis, positional amino acid property mapping, and k-mer analysis enhance functional interpretation.

- Machine learning (ML) applications: Integration with Trex, Ibex, and ImmApex expands predictive modeling capabilities, particularly for immunology-related deep learning applications.

Overall, scRepertoire 2 is a well-designed computational tool for immune repertoire analysis, introducing significant enhancements in data compatibility, performance, and machine learning applications. However, critical aspects such as comparative benchmarking, validation of functional predictions, batch effect correction, and scalability testing remain underexplored. Addressing these limitations will further strengthen its utility in high-throughput immunogenomics.

I have some listed concerns & suggested improvements

1. Design and Implementation / Workflow:

The workflow efficiently integrates immune repertoire data, but it lacks explicit benchmarking against existing tools such as Immunarch, scIR, or TCRgrapher to demonstrate comparative performance improvements.

- Suggested Change: Include quantitative benchmarking (e.g., accuracy of clonal assignments, runtime comparisons).

2. Expanded Data Compatibility:

The newly introduced `loadContigs()` function automatically detects formats, which is a strong feature. However, the details of how misclassified formats are handled are not described.

- Suggested Change: Provide error-handling mechanisms (e.g., warnings for ambiguous format detection).

3. Performance Optimizations:

While C++ acceleration improves runtime efficiency, the scalability for large repertoires (>1M cells) is not explicitly tested.

- Suggested Change: Include real-world benchmarking for large single-cell datasets, preferably using public datasets from immune cell atlases.

4. Clonal Diversity Analysis:

While rarefaction analysis is robust, it lacks a discussion on statistical variability (e.g., how confidence intervals are derived).

-Suggested Change: Report statistical uncertainties (e.g., confidence intervals, bootstrap variability).

5. ML Applications:

The ML component is promising, but it lacks clarity on training data availability and whether models are trained on independent datasets.

- Suggested Change: Provide details on model generalizability, including cross-validation performance on independent cohorts.

**Have the authors made all data and (if applicable) computational code underlying the findings in their manuscript fully available?**

Reviewer #1: Yes

Reviewer #2: Yes

Reviewer #3: Yes

Reviewer #4: Yes

PLOS authors have the option to publish the peer review history of their article (what does this mean? ). If published, this will include your full peer review and any attached files.

**Do you want your identity to be public for this peer review?** For information about this choice, including consent withdrawal, please see our Privacy Policy .

Reviewer #1: **Yes: ** Isar Nassiri

Reviewer #2: No

Reviewer #3: **Yes: ** Junyue Cao

Reviewer #4: No

**Figure resubmission:**
---

## [Decision Letter · Decision Letter 1]

Dear Dr Borcherding,

We are pleased to inform you that your manuscript 'scRepertoire 2: Enhanced and Efficient Toolkit for Single-Cell Immune Profiling' has been provisionally accepted for publication in PLOS Computational Biology.

Best regards,

Pramod Shinde

Guest Editor

PLOS Computational Biology

Amber Smith

Section Editor

PLOS Computational Biology

Thank you to the authors for thoroughly addressing the reviewers' comments. The revised manuscript satisfactorily resolves the concerns raised during peer review. I am pleased to recommend the manuscript for acceptance.

Reviewer's Responses to Questions

**Comments to the Authors:**

Reviewer #1: I appreciate the authors' diligent work on the revisions for the manuscript. Upon reviewing their responses and the revised manuscript, I am convinced that all my concerns have been addressed. I suggest that the manuscript is now suitable for publication.

Reviewer #2: Authors have addressed all the comments. Manuscript can be accepted.

Reviewer #3: The author has addressed my concerns and suggestions. The addition of quantitative benchmarking, demonstration of ML packages, report on scalability, and the discussion of alternative approaches have significantly improved the manuscript. I am satisfied with the revision and recommend this manuscript for publication.

Reviewer #4: As the comments are fairly addressed, I would recommend the acceptance of the article for publication.

**Have the authors made all data and (if applicable) computational code underlying the findings in their manuscript fully available?**

Reviewer #1: Yes

Reviewer #2: Yes

Reviewer #3: Yes

Reviewer #4: Yes

PLOS authors have the option to publish the peer review history of their article (what does this mean? ). If published, this will include your full peer review and any attached files.

**Do you want your identity to be public for this peer review?** For information about this choice, including consent withdrawal, please see our Privacy Policy .

Reviewer #1: No

Reviewer #2: **Yes: ** Ankush Bansal

Reviewer #3: No

Reviewer #4: **Yes: ** Aquib Ehtram

---

## [Editor Report · Acceptance letter]

PCOMPBIOL-D-24-02249R1

scRepertoire 2: Enhanced and Efficient Toolkit for Single-Cell Immune Profiling

Dear Dr Borcherding,

I am pleased to inform you that your manuscript has been formally accepted for publication in PLOS Computational Biology. Your manuscript is now with our production department and you will be notified of the publication date in due course.

With kind regards,

Lilla Horvath
